# Self-Healable, Strong, and Tough Polyurethane Elastomer Enabled by Carbamate-Containing Chain Extenders Derived from Ethyl Carbonate

**DOI:** 10.3390/polym14091673

**Published:** 2022-04-20

**Authors:** Pengcheng Yi, Jingrong Chen, Junyao Chang, Junbo Wang, Ying Lei, Ruobing Jing, Xingjiang Liu, Ailing Sun, Liuhe Wei, Yuhan Li

**Affiliations:** Zhengzhou Key Laboratory of Elastic Sealing Materials, College of Chemistry and Green Catalysis Center, Zhengzhou University, Zhengzhou 450001, China; yipengcheng@gs.zzu.edu.cn (P.Y.); cjrkate@stu.zzu.edu.cn (J.C.); cjyailyh1212@stu.zzu.edu.cn (J.C.); 202023000516@stu.zzu.edu.cn (J.W.); 202023000403@stu.zzu.edu.cn (Y.L.); 202023000105@stu.zzu.edu.cn (R.J.); xingjiangliu@zzu.edu.cn (X.L.); ailingsun@zzu.edu.cn (A.S.)

**Keywords:** carbamate-containing chain extenders, high strength, high self-healing ability, adhesive properties

## Abstract

Commercial diol chain extenders generally could only form two urethane bonds, while abundant hydrogen bonds were required to construct self-healing thermoplastic polyurethane elastomers (TPU). Herein, two diol chain extenders bis(2-hydroxyethyl) (1,3-pheny-lene-bis-(methylene)) dicarbamate (BDM) and bis(2-hydroxyethyl) (methylenebis(cyclohexane-4,1-diy-l)) dicarbamate (BDH), containing two carbamate groups were successfully synthesized through the ring-opening reaction of ethylene carbonate (EC) with 1,3-benzenedimetha-namine (MX-DA) and 4, 4′-diaminodicyclohexylmethane (HMDA). The two chain extenders were applied to successfully achieve both high strength and high self-healing ability. The BDM-1.7 and BDH-1.7 elastomers had high comprehensive self-healing efficiency (100%, 95%) after heated treatment at 60 °C, and exhibited exceptional comprehensive mechanical performances in tensile strength (20.6 ± 1.3 MPa, 37.1 ± 1.7 MPa), toughness (83.5 ± 2.0 MJ/m^3^, 118.8 ± 5.1 MJ/m^3^), puncture resistance (196.0 mJ, 626.0 mJ), and adhesion (4.6 MPa, 4.8 MPa). The peculiar mechanical and self-healing properties of TPUs originated from the coexisting short and long hard segments, strain-induced crystallization (SIC). The two elastomers with excellent properties could be applied to engineering-grade fields such as commercial sealants, adhesives, and so on.

## 1. Introduction

Artificial polymer materials are often unusable or have their service life greatly reduced after physical damage, resulting in environmental pollution and a great waste of resources [1,2,3]. Studies have found that almost all natural organisms can spontaneously heal themselves after physical injury, greatly increasing their survival capacity and life span [4,5]. The salamander, for example, has an amazing ability of self-healing and can heal almost any organ [6]. Based on the inspiration that living things can heal themselves, people have been devoted efforts to the research of self-healing polymer materials in recent decades, which can increase the service life of materials, preserve their stability, reduce the maintenance cost, and further relieve the pressure of resource waste and environmental pollution [7,8].

In terms of the healing mechanism, the healing modes of polymer materials can be divided into external healing and internal healing. The latter has attracted more attention given its repeatable self-healing and low influence on material properties. Internal healing introduces dynamic reversible covalent bond [9,10,11,12,13,14,15] or non-covalent bond [16,17,18,19] into the polymer chain to achieve self-healing more efficiently. Internally healed polymer materials are widely used considering their high-level applicability and mass production [20,21,22] while polyurethane elastomer has become one of the most widely used polymer materials by virtue of its excellent properties [23,24,25]. Self-healing polyurethane materials are widely applied in electronic skin, intelligent sensors, biomedical materials, and many other areas, where the mechanical properties of self-healing polyurethane materials are generally poor. However, in some sealants and commercial coatings [26], polyurethane materials are required to be self-healing and have high requirements for mechanical properties such as strength and toughness. Therefore, it is of great significance to develop self-healing polyurethane elastomers with a strong self-healing ability, as well as excellent strength and toughness performance in these fields.

It is undeniable that the mobility of chain segments is the most critical factor in promoting self-healing in the internal healing mode of polyurethane elastomers [2,27]. High self-healing efficiency of elastomers has poor mechanical properties in most cases [28,29,30,31]. The strength rise of polyurethane elastomer is generally accompanied by the content rise of hard segments, forming more physical cross-linking points between soft segments. It is not conducive to self-healing when the movement of the chain segments is limited. It composes a paradox that improvement in mechanical properties is followed by a decline in self-healing efficiency. Therefore, it is still challenging to find rational molecular chain design and innovative strategies to resolve such a contradictory relation [4,32].

An innovative strategy was proposed to develop polyurethane elastomers with high strength self-healing ability, and solve the contradiction between mechanical properties and the self-healing ability of TPU. Polytetramethylene ether glycol (PTMEG) was selected as the polymeric diol because its SIC would produce a mechano-response self-strengthening effect, which will further synchronously strengthen and toughen the elastomer [33,34,35,36]. Carbamate-containing chain extenders, BDM and BDH, were successfully synthesized from inexpensive raw materials. BDM and BDH were incorporated into the polyurethane backbone, with high steric hindrance benzene and cyclohexane. These large steric groups made the whole hard segments loosely-stacked, which rendered the chain segments strong fluidity, and endowed the elastomers with high self-healing performance. The special design of the long and short hard segments in BDM and BDH elastomers facilitated the formation of SIC and introduced suitable amounts of hydrogen bonds, which granted TPU high mechanical properties.

## 2. Experimental

### 2.1. Materials

PTMEG (M_n_ = 1000 g/mol, f = 2), dibutyltin dilaurate (DBTDL), and ethylene glycol (EG) were purchased from Aladdin (Shanghai, China). EC, MXDA, HMDA and 4,4′-dicyclohexylmethane diisocyanate (HMDI) were from Macklin (Shanghai, China). None of the above chemicals were further purified. Both toluene and tetrahydrofuran (THF) were redistilled by CaH_2_ for use.

### 2.2. Synthesis of BDM, BDH

BDM and BDH were synthesized from EC, MXDA, and HMDA via ring-opening reactions. At room temperature, EC and MXDA or HMDA were mixed thoroughly in a 250-mL flask in a molar ratio of 2.05/1 (MXDA strongly reacting with EC and should be added in batches). The temperature of the reaction system was raised to 120 °C, when the reaction would last for 12 h. To ensure the full reaction of MXDA and HMDA, the amount of EC was slightly excessive during such a process. After the reaction, the product was correspondingly dissolved, recrystallized, filtered and dried to get BDM and BDH powder (Appendix A). FTIR (Appendix A) and ^1^HNMR (Appendix A) proved the successful synthesis of BDM and BDH.

### 2.3. Synthesis of Polyurethane Elastomers Using BDM and BDH as Chain Extenders

Two types of linear polyurethane elastomers with different R values (defined as the molar ratio of isocyanate group to hydroxyl group in PTMEG[NCO]/[OH]) were synthesized via the same process. Take the synthesis process of BDM-2.0 elastomer as an example: firstly, PTMEG (20.0 g, 20.0 mmol) was added into a 250-mL four-neck flask with a mechanical stirrer and a mercury thermometer, then heated at 120 °C with an electric heating jacket for 1 h under vacuum to remove the residual moisture in PTMEG, and cooled to 60 °C after that. HMDI (10.5 g, 40.0 mmol) was added into the flask and stirred at 80 °C for 1 h under argon gas protection. Then, DBTDL (about 2.5 mg, 4.0 × 10^−3^ mmol) was added and reacted at 80 °C for 2 h to obtain the prepolymer. After that, BDM (6.2 g, 20.0 mmol) was added, and then 70 mL of THF was added to the system and stirred at 60 °C for about 0.5 h (to adjust the viscosity and dissolve BDM powder). After BDM was completely dissolved, 100 mL of toluene solution and DBTDL (25.3 mg, 4.0 × 10^−2^ mmol) were added to react at 80 °C for 2 h, so that all NCO groups could be completely consumed. The product was poured into the prepared rectangular mold, dried at 80 °C for 48 h, and then dried in a vacuum drying oven at 80 °C for 24 h to remove the residual solvent. The obtained polyurethane film (1 ± 0.2 mm) was labeled as BDM-2.0. With EG as chain extender and R = 1.7, the polymer was synthesized and labeled as EG-1.7.

### 2.4. Tensile Tests

Mechanical properties including puncture, tensile strength, elongation at break and toughness were tested by a 500 N load sensor tension tester (TH-8203A, TOPHUNG Inc., Suzhou, China): while dumbbell (DIN 53504, Type S1) samples were cut from the prepared elastomer film (1 ± 0.2 mm) and tensile tests were also performed at a constant rate of 100 mm/min at room temperature. A steel needle was mounted on the same tensile testing machine, and the puncture test was carried out at a constant speed of 50 mm/min until about 0.5 mm of the film was penetrated. Puncture energy was calculated by integral of force–displacement curve. At room temperature, the spline was stretched to 300% strain at a constant rate of 50 mm/min. After five cycles of continuous loading and unloading, the cyclic stretching curve could be obtained after staying at room temperature for 12 h and after one recirculating. Self-healing performance was evaluated by tensile test according to the following methods: at least five dumbbell samples were cut with a razor in the middle and splice immediately. The tensile test was carried out after a certain period of treatment at different temperatures. Self-healing efficiency (S.E.) was calculated by tensile strength, elongation and toughness: (P_healed_/P_original_) × 100%, where P is tensile strength, elongation at break or toughness.

### 2.5. Adhesion Tests

Tensile shear strength tests were conducted by a 100 KN load sensor tension tester (TH-8100A, TOPHUNG Inc., Suzhou, China) with a 100 mm × 25 mm × 0.8 mm steel plate lap whose, lap area was 25 mm × 12.5 mm. Not less than 5 samples were heated in the oven at 80 °C and 150 °C for a certain time, then adjusted 24 h to the equilibrium state at room temperature and tested at a constant speed of 50 mm/min. Tensile shear strength τ = F/B (unit: MPa), where F is the maximal shear failure load (N) and B is the lap area (mm^2^).

### 2.6. Characterizations

The film samples were under FTIR test in the attenuated total reflection (ATR) mode using a Bruker ALPHA II (Baden, Württemberg, Germany) Fourier transform infrared spectrometer. The resolution was 4 cm^−1^, the amount of scanning, 32; and the spectral scanning, ranging from 4000 to 500 cm^−1^. Nicolet 6700 produced by Thermo Electron Company (Waltham, MA, USA) in the United States was used for variable-temperature FTIR experiments. The test temperature was from 25 to 105 °C while the interval temperature was 10 °C, and the residence time was 3 min. The ^1^HNMR test was performed by Bruker Avance 400 MHz (Baden, Württemberg, Germany), DMSO and CDCl_3_ as solvent. Gel permeation chromatography (GPC, Agilent LC1200, Agilent Technologies, Santa Clara, CA, USA) evaluated molecular weight and polydispersion index with THF as a mobile phase. DSC measurements were carried out using TA-DSC250 (TA Instruments, New Castle, DE, USA) at a heating rate of 10 °C/min in a nitrogen atmosphere. During the first operation, the temperature dropped from to −50 °C for 3 min, and then increased to 150 °C and kept for 3 min. For the second run, the temperature was lowered from 150 to −50 °C and finally to 150 °C. Each sample was scanned at the rate of 10 °C/min. Polarization optical microscope (POM, Olympus BX61, Olympus, Tokyo, Japan) was used to observe the surface self-healing of the samples. Small Angle X-ray scattering (SAXS) was performed by Bruker-Anaostar (Bruker, Karlsruhe, Germany) at 50 KV and 0.6 mA. The radiation source was Cu Kα, λ= 1.54056 Å, and the distance between the sample and the detector was 450 mm. Rheological properties were tested using a rotational rheometer (Malvern, London, UK) with a 20 mm circular membrane. Frequency sweeping was at a strain amplitude of 0.1%, in the range of 0.1–100 Hz (90 °C). Temperature sweeping was in the range of 25 to 150 °C.

## 3. Results and Discussion

### 3.1. Molecular Design

The two chain extenders with carbamate groups, BDM and BDH, were synthesized by ring-opening reaction (Figure 1a): given that their benzene ring and two cyclohexyl groups, which possessed high steric resistance and loosely-stacked hard segments structure, could be formed in the elastomer, their carbamate groups within the synthesized chain extenders also introduce further more hydrogen bonds. The special structure of these two chain extenders laid a foundation for the preparation of high mechanical properties and high self-healing elastomers. BDM and BDH were skillfully introduced into TPU by the two-step prepolymer method. In the first step, HMDI reacted with PTMEG to generate prepolymers and form short hard segments. The specific feed ratio of HMDI/PTMEG was showed in Appendix A. In the second step, BDM and BDH, as chain extenders, formed loosely-stacked short hard segments and long hard segments (Figure 1b,c). FTIR (Appendix A), ^1^HNMR (Appendix A) and GPC (Appendix A) proved that the above methods could successfully synthesize BDM-based, BDH-based and EG-1.7 thermoplastic polyurethane elastomers.

Polyurethane elastomer contained polar groups such as carbamate. Due to its large cohesion, hydrogen bonds could be formed between molecules and spontaneously aggregate into a hard phase region, while the chain segments of polyesters or polyether diols with weak polarity aggregated into a soft phase region. The hard phase and soft phase were thermodynamically incompatible, resulting in a special phase morphology-microscopic phase separation. The special phase morphology of BDM-1.7 and BDH-1.7 elastomers was characterized by SAXS. As showed in Figure 2a, an obvious scattering peak existed inevitably due to different electron densities of the soft phase and the hard phase. The results showed that there was a microphase separation structure between these two phases, when the hard phase acted as physical cross-linking points within the soft phase, which enhances BDM-1.7 and BDH-1.7 elastomer strength.

BDM-1.7 and BDH-1.7 elastomers were tested by variable-temperature FTIR, and the test results were shown in Figure 2b,c. The characteristic peaks of the C=O group in BDM-1.7 and BDH-1.7 were subjected to the same trend with the temperature. BDM-1.7 spectrum was taken as an example to explain that, from 25 to 105 °C, the intensity of C=O characteristic peak (1716 cm^−1^) in the free state of amide I gradually increased, while the trend of C=O characteristic peak (1705 cm^−1^) in the hydrogen bond association state of amide I performed opposite, indicating the dynamic reversibility of hydrogen bonds in BDM-based and BDH-based elastomers with the increase of temperature.

### 3.2. Mechanical Properties

Next, the mechanical properties of BDM-based and BDH-based elastomers were discussed from the aspects of tensile, puncture, and cyclic tensile experiment, showing that the hard phase possessed the advantage of both short and long hard segment structure. Stress-strain curves of the BDM-based and BDH-based elastomers were shown in Figure 2d,e, and specific data were shown in Appendix A. It was obvious that the elongation at break decreased with increasing R values while the ultimate tensile strength augmented with increasing R values. It also showed that the content of the long hard segment greatly influenced strain hardening. The tensile strength of BDM-1.7 and BDH-1.7 elastomers was found to be significantly increased, due to the SIC effect of soft phase PTMEG chain in synthetic TPU elastomers, which played a positive role in the strengthening and toughening. The tensile strength of BDM-1.4 and BDH-1.1 was only 0.61 ± 0.01 MPa and 0.19 ± 0.03 MPa, respectively. Although the molecular weight of synthesized TPU was high, the low content of long hard segments made the amounts of physical cross-linking points too small (Appendix A), and the density was not enough to maintain the integrity of polyurethane network structure during the tensile process. The chain segment would be broken when the strain was low. The tensile strength of BDH-1.7 and BDH-1.7 elastomers was 20.55 ± 1.3 MPa and 37.07 ± 1.7 MPa, and the elongation at break was 1667 ± 56% and 1128 ± 53%, respectively. The special design of the long and short hard segments in BDM-1.7 and BDH-1.7 elastomers facilitates the formation of SIC and introduced suitable amounts of hydrogen bonds. At this time, the elastomer possessed a high content of long hard segments, forming a large number of physical cross-linking points, so that the whole polymer network could restrict chain migration in the tensile process, resulting in higher mechanical properties.

**Figure 2 polymers-14-01673-f002:**
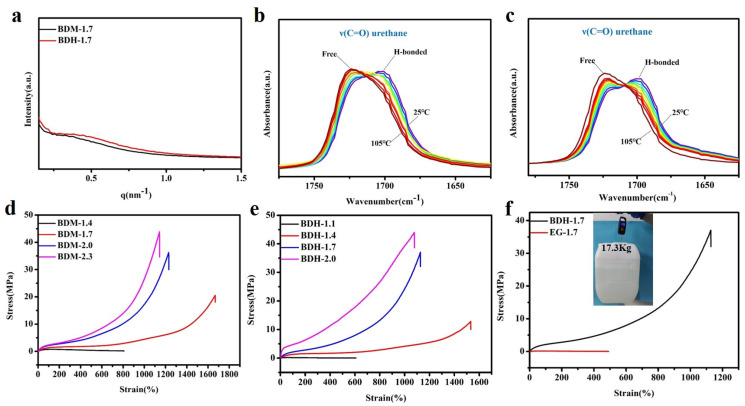
(**a**) SAXS results of BDM-1.7, BDH-1.7 elastomers; (**b**) Variable-temperature FTIR spectra of BDM-1.7; (**c**) Variable-temperature FTIR spectra of BDH-1.7; (**d**) Stress-strain curves of BDM-based elastomers; (**e**) Stress-strain curves of BDH-based elastomers; (**f**) Stress-strain curves of BDH-1.7, EG-1.7 elastomers and a tape of BDH-1.7 elastomer lifting a heavy bucket.

To intuitively highlight the advantages of the special structures of BDM and BDH in enhancing the mechanical properties of polyurethane elastomers, EG-1.7 elastomer was synthesized, with EG, which had no carbamates and a low steric resistance, as a chain extender. Figure 2f showed the stress-strain curves of BDH-1.7 and EG-1.7 elastomers. EG-1.7 had a high molecular weight (Appendix A), but its tensile strength was only 0.13 ± 0.02 MPa, since the content of the hard segments of EG-1.7 elastomer was low, among which, the content of long hard segment (Figure 1d) was particularly low (Appendix A). At this time, the amounts of physical cross-linking points were too small, and the density of physical cross-linking points was not enough to maintain the integrity of the polyurethane network structure during the tensile process. It would be broken when the strain was low, since BDH-1.7 elastomer possessed a high content of long hard segments that provided more physical cross-linking points and contained rigid cyclohexyl structure and carbamate groups which mattered considerably in enhancing the mechanical properties, the tensile strength of BDH-1.7 elastomer was 37.07 ± 1.7 MPa, much higher than EG-1.7. It was found that the special structure of BDM and BDH contributed for improving the mechanical properties of polyurethane elastomers. Benefited from strain hardening, BDH-1.7 elastomer is provided with a high strength and BDH-1.7 tape (thickness of 1.3 mm) that could lift a bucket weighing 17.3 kg (Figure 2f).

Puncture resistance is defined as material ability to avoid the puncture of a sharp needle. BDM-based and BDH-based elastomers possess excellent puncture resistance. Their puncture experiments on the tensile testing machines could obtain force-displacement curves (Figure 3a,c). It could be seen from the above data that with the increased of R value, the maximum puncture force of BDM-based and BDH-based elastomers increased, while the maximum puncture displacement decreased. When the R value increased, the number of hydrogen bonds in the elastomer increased correspondingly, making it difficult for the molecular chain to move. At this time, when the steel needle penetrated the film, it presented a higher penetration force and higher penetration resistance. The maximal puncture force of BDH-2.0 was 39.2 N, slightly lower than that of BDH-1.7. This might result from the fact that the hard segment content was too high when the R value of BDH elastomer was 2.0, which gave rise to too many hydrogen bonds that limited the chain movement. This puncture resistance of BDH-2.0 elastomer was weakened, resulting in a short maximal puncture displacement and failure to achieve the desired puncture force. The results showed that with the increase of R value, the rise of BDM and BDH content was the key to improving puncture resistance. Puncture energy could be obtained by integrating the force-displacement curves (Figure 3b,d). It could intuitively reflect the energy dissipation capacity during puncture deformation. The maximal puncture energy of the BDM-1.7 elastomer was 196.0 mJ, and that of the BDH-1.7 elastomer was 627.0 mJ. BDM-1.7 and BDH-1.7 elastomers were subjected to cyclic tensile test (Figure 3e,f) for their resilience. After staying at room temperature for 12 h, the dissociated hydrogen bonds were re-associated, and the last cycle curve was close to the first one, indicating that BDM-1.7 and BDH-1.7 elastomers have high resilience which could also prove the dynamic reversibility of hydrogen bonds. It was also found that BDM-1.7 and BDH-1.7 elastomers had a high strength, high puncture resistance and high resilience, laying a good foundation for their practical application.

### 3.3. Self-Healing Properties

According to the DSC results (Figure 4a), the T_g_ of hard segment of BDM-1.7 and BDH-1.7 elastomers were 30.4 and 29.8 °C, respectively. Theoretically, when the temperature was higher than the T_g_ temperature, these two kinds of elastomers were in high elastic state and had certain self-healing ability. Both DSC and variable-temperature FTIR results indicate that BDM-based and BDH-based elastomers may have self-healing ability after heating. To confirm the self-healing ability of BDM-based and BDH-based elastomers after heating, the elastomer specimens of BDM-1.7 and BDH-1.7 were cut and spliced together immediately. After heating treatment, the BDM-1.7 and BDH-1.7 elastomers were subjected to suspension experiments (Appendix A). It was found that these two types of elastomers could successfully withstand a weight of 500 g after the healing with a short period of heat treatment. The results showed that BDM-1.7, BDH-1.7 elastomer did have a strong self-healing ability after heat treatment. The scratches before and after the self-healing of BDM-1.7 and BDH-1.7 elastomers were observed with an optical microscope (Figure 4b,c), and the self-healing ability of BDM-1.7 and BDH-1.7 elastomers could be more clearly marked. It could be seen that scratches would almost disappear completely after heating at 60 °C for 24 h. It further indicated that BDM-1.7 and BDH-1.7 elastomers possessed a strong self-healing ability.

To further quantify the self-healing efficiency of BDM-based and BDH-based elastomers, specimens of BDM-1.7 and BDH-1.7 were cut in the middle and quickly spliced together. They were heat-treated in ovens at 25, 40, and 60 °C, and tensile tests were conducted to obtain the corresponding stress-strain curves (Figure 4d,e). Herein, considering the strain hardening of the SIC effect, it should be that self-healing efficiency was synthetically evaluated by the repairing of tensile strength, elongation and toughness instead of an individual one. According to their self-healing data, the self-healing efficiency of the elastomer increased with the increase of temperature. When BDM-1.7 and BDH-1.7 elastomers were treated at 60 °C for 24 h, the self-healing efficiency (the tensile strength, elongation at break and toughness) of BDM-1.7 elastomers reached nearly 100%, and the self-healing efficiency(the tensile strength, elongation at break and toughness) of BDH-1.7 elastomers reached more than 95% (Appendix A). The results showed that under these conditions, the special phase morphology of long and short hard segment in elastomer and the design of BDM and BDH structure endowed the elastomer with high mechanical properties and the chain segment highly fluidity. Under the heating condition, the fluidity of the chain segment of the elastomer increased, and new hydrogen bonds were rapidly formed at the scratches, making the scratches remerge quickly. When healing time increased, the self-healing ability was further enhanced, and the mechanical properties were basically restored. BDM-1.7 and BDH-1.7 elastomers also possessed a high toughness, reaching 85.5 ± 2.0 MJ/m^3^ and 118.8 ± 5.1 MJ/m^3^, respectively. The strength and toughness of BDM-2.0, BDM-2.3, and BDH-2.0 with a higher R value were greatly improved, but the self-healing ability was decreased instead, indicating that the mechanical properties and self-healing efficiency of BDM-based and BDH-based elastomers could be regulated by changing the R value. BDH-1.7 was heated at 60 °C for 4, 8, and 12 h to obtain the stress-strain curve (Figure 4f). The results showed that the self-healing ability of BDH-1.7 elastomer was significantly improved with the prolongation of the healing time.

To further analyze the self-healing mechanism of BDM-1.7 and BDH-1.7 elastomers, rheological tests were therefore carried out, and their rheological curves were plotted at 90 °C (Figure 5a,c). The relaxation time (τ_f_) of the whole chain flow was calculated from the reciprocal of G′=G″ used to indicate the size of chain mobility. The relaxation time of BDM-1.7 and BDH-1.7 was 34 and 18 s, respectively, extremely short, which means that BDM-1.7 and BDH-1.7 both possessed an extremely high chain mobility at 90 °C. According to the results of the self-healing experiment, BDM-1.7 and BDH-1.7 elastomers had a higher self-healing efficiency at 60 °C after the heat treatment of 24 h. These results indicated that the hard segments of BDM-1.7 and BDH-1.7 elastomers were loosely-stacked and had a high chain mobility at 60 °C. It also represented a rapid rearrangement of their hard segments after heating. The increase of self-healing time and temperature improved the corresponding efficiency, which could be obviously explained by the increase of segment mobility of BDM-based and BDH-based elastomers.

### 3.4. Adhesion Properties

In the above tests, BDM-based and BDH-based elastomers were found to be soft and viscous when heated, similar to adhesives. The adhesion properties of these two types of elastomers were tested. They were heated in an oven at 80 °C for 8 h and 150 °C for 0.5 h, respectively, and then adjusted at room temperature for 24 h for the tensile shear test. The shear strength-displacement curves (Figure 6a,c) and shear strength (Figure 6b,d) of BDM elastomer at different temperatures were obtained, and the shear strength-displacement curves (Figure 6g,i) and shear strength (Figure 6h,j) of BDH elastomer at different temperatures were obtained. The adhesion performance of BDM-based and BDH-based elastomers at 80 °C was much weaker than that at 150 °C, since their melting temperature was higher. It could be observed in the temperature scanning curves (Figure 5b,d) that the melting temperatures were 139.6 and 126.2 °C, respectively. The elastomers become soft when treated at 80 °C and could not nicely adhere to the substrate, so the shear strength was low. When heated at 150 °C for 0.5 h, large amounts of hydrogen bonds could interact with the surface of the substrate, and the hydrogen bonds in the material could also achieve rapid dynamic exchange and recombination to obtain higher cohesive energy. It could nicely adhere with the substrate, so it had a higher shear strength. With the increase of R value, the adhesion properties of BDM-based and BDH-based elastomers increase. Because of the increase of R value, the content of hard segment increased continuously, increasing the number of hydrogen bonds, and the elastomer had a strong interaction with the substrate after heating. BDM-1.7 and BDH-1.7 elastomers possessed relatively excellent adhesion properties. The shear strength of BDM-1.7 and BDH-1.7 elastomers was 1.37 and 2.29 MPa, respectively, when heated at 80 °C for 8 h. When heated at 150 °C for 0.5 h, the shear strength was 4.59 and 4.82 MPa, respectively, which were comparable with some commercially available adhesives. As R value continues to increase, the shear strength of BDM-2.3 and BDH-2.0 elastomers became the largest, reaching an astonishing 12.91 and 11.32 MPa when heated at 150 °C for 0.5 h. The shear strength reached 5.70 and 4.55 MPa at 80 °C for 8 h, respectively. The adhesion properties of BDM-2.3 and BDH-2.0 elastomers with maximal shear strength were tested at 80 °C for different heating times, and the shear strength-displacement curves (Figure 6e,k) and shear strength (Figure 6f,l) of BDM-2.3 and BDH-2.0 elastomers were obtained at 80 °C for different heating times. It could be observed that the highest shear strength was achieved when BDH-2.0 elastomers were heated for 2 h at 80 °C. The shear strength of BDM-2.3 reached the maximal after heated for 8 h. These results indicated that elastomers with different adhesion properties could be obtained by adjusting the R value, temperature and heating time. The elastomer with high adhesion performance could be widely used in electronic assembly, aerospace, and other fields.

## 4. Conclusions

In summary, two chain extenders, BDM and BDH, were synthesized by ring-opening reaction with the five-membered EC. BDM and BDH as chain extenders were adopted and the special structure of short hard and long hard segments was skillfully introduced by regulating the R value. This special structure facilitated the formation of SIC and introduced suitable amounts of hydrogen bonds, which contributed to the improvement of their mechanical properties. Due to the special structure of BDM and BDH, the synthesized polyurethane elastomer was proven to possess the loosely-stacked hard segment structure, beneficial for the improvement of their self-healing ability. Two polyurethane elastomers with excellent comprehensive properties were synthesized. The results showed that the tensile strength of the BDM-1.7 and BDH-1.7 elastomers was 20.6 ± 1.3 MPa and 37.1 ± 1.7 MPa, the toughness, 83.5 ± 2.0 MJ/m^3^ and 118.8 ± 5.1 MJ/m^3^, and the puncture energy, 196.0 and 627.0 mJ, respectively. Under the condition of 60 °C for 24 h, the comprehensive self-healing efficiency of BDM-1.7 elastomer reached almost 100%, and that of BDH-1.7 elastomer is above 95%. In terms of adhesive properties, the shear strength of BDM-1.7 and BDH-1.7 elastomers was 4.59 and 4.82 MPa, respectively, after being treated at 150 °C for 0.5 h. Based on all of these, this study provides an innovative strategy for further exploration on elastomers with high mechanical properties and excellent self-healing ability.

## Figures and Tables

**Figure 1 polymers-14-01673-f001:**
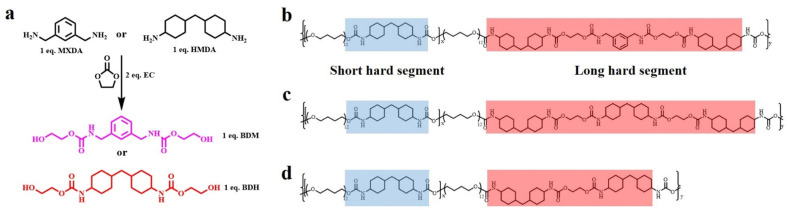
(**a**) Synthetic routes of BDM and BDH; (**b**) Molecular structure of BDM-1.7 elastomers embedded with short and long hard segments; (**c**) Molecular structure of BDH-1.7 elastomers embedded with short and long hard segments; (**d**) Molecular structure of EG-1.7 elastomers embedded with short and long hard segments.

**Figure 3 polymers-14-01673-f003:**
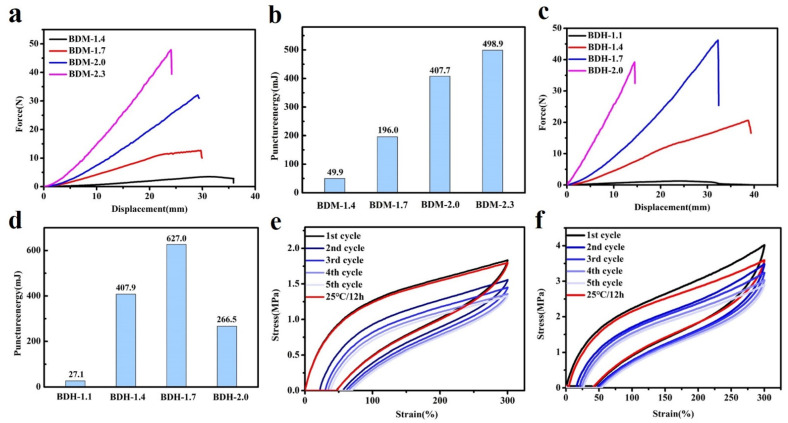
(**a**) Force-displacement curves for BDM-based elastomer; (**b**) Puncture energies of BDM-based elastomers calculated from force-displacement curves; (**c**) Force-displacement curves for BDH-based elastomer; (**d**) Puncture energies of BDH-based elastomers calculated from force-displacement curves; (**e**,**f**) Cyclic test curves of BDM-1.7, BDH-1.7 elastomer.

**Figure 4 polymers-14-01673-f004:**
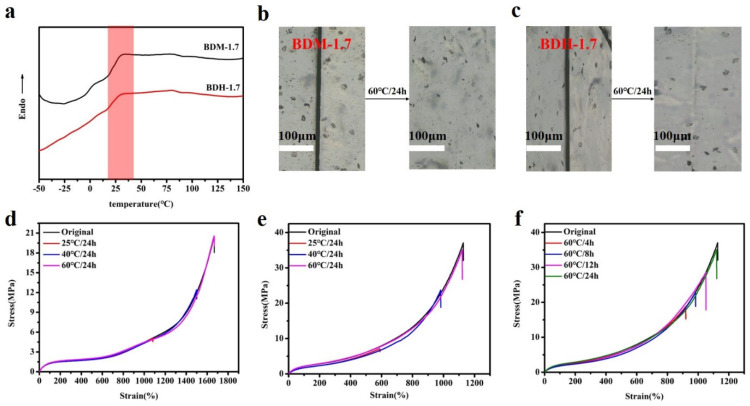
(**a**) DSC curves of BDM-1.7, BDH-1.7 elastomers; (**b**) and (**c**) BDM-1.7, BDH-1.7 elastomers self-healing at 60 °C for 24 h, change of scratches; (**d**) and (**e**) Stress-strain curves of BDM-1.7, BDH-1.7 elastomers self-healing at 25, 40, and 60 °C for 24 h; (**f**) Stress-strain curves for BDH-1.7 elastomers self-healing at 60 °C for 4, 8, 12, and 24 h.

**Figure 5 polymers-14-01673-f005:**
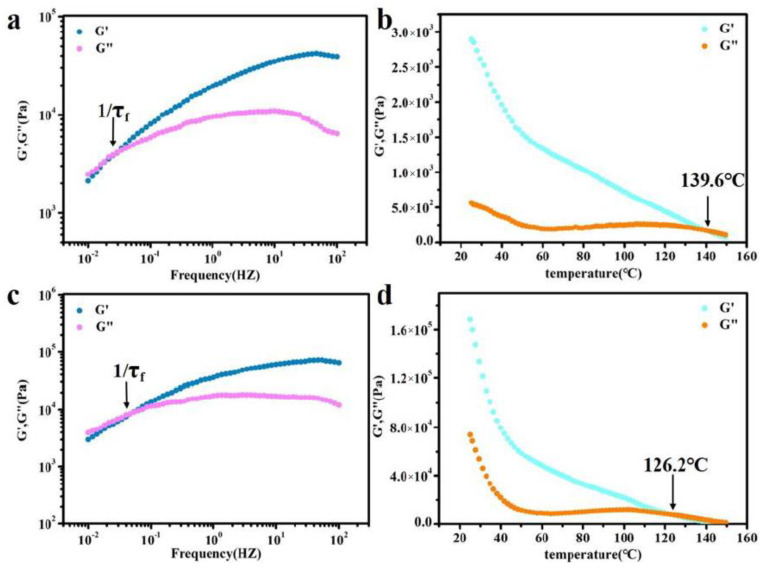
(**a**) Rheological curves of BDM-1.7 elastomers (90 °C) for deducing relaxation time of chain flow transition τ_f_; (**b**) Temperature sweep of rheological testing for the BDM-1.7 elastomer; (**c**) Rheological curves of BDH-1.7, elastomers (90 °C) for deducing relaxation time of chain flow transition τ_f_; (**d**) Temperature sweep of rheological testing for the BDH-1.7 elastomer.

**Figure 6 polymers-14-01673-f006:**
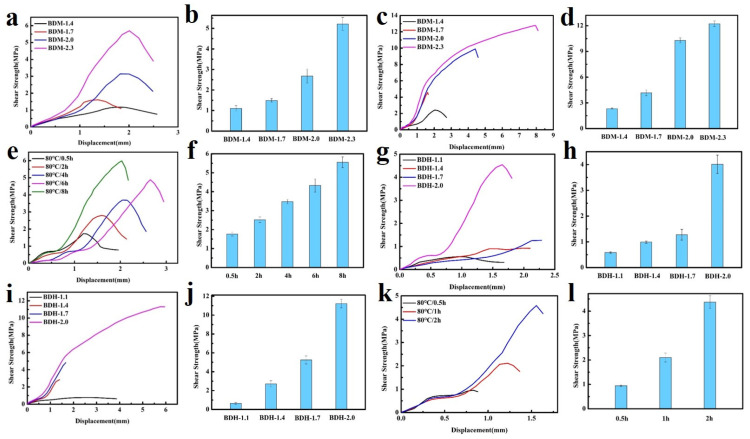
(**a**) Shear strength-displacement curves of BDM-based elastomer at 80 °C for 8 h; (**b**) Shear strength of BDM-based elastomers at 80 °C for 8 h; (**c**) Shear strength-displacement curves of BDM-based elastomer at 150 °C for 0.5 h; (**d**) Shear strength of BDM-based elastomers at 150 °C for 0.5 h; (**e**) Shear strength-displacement curve of BDM-2.3 elastomers at 80 °C for 0.5, 2, 4, 6, and 8 h; (**f**) Shear strength of BDM-2.3 elastomers at 80 °C for 0.5, 2, 4, 6, and 8 h; (**g**) Shear strength-displacement curves of BDH-based elastomer at 80 °C for 8 h; (**h**) Shear strength of BDH-based elastomers at 80 °C for 8 h; (**i**) Shear strength-displacement curves of BDH-based elastomer at 150 °C for 0.5 h; (**j**) Shear strength of BDH-based elastomers at 150 °C for 0.5 h; (**k**) Shear strength-displacement curve of BDH-2.0 elastomers at 80 °C for 0.5, 1, and 2 h; (**l**) Shear strength of BDH-2.0 elastomers at 80 °C for 0.5, 1, and 2 h.

## Data Availability

Not applicable.

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
