# Peer review of "Self-Healable, Strong, and Tough Polyurethane Elastomer Enabled by Carbamate-Containing Chain Extenders Derived from Ethyl Carbonate"

_polymers, 2022, doi:10.3390/polym14091673_

Round 1

Reviewer 1 Report

In this manuscript, the authors reported that they synthesized two diol-chain extenders (BDM and BDH) containing two carbamate groups through the ring-opening reaction of EC with MXDA and HMDA. Further, two elastomers with high strength and high self-healing ability were developed by applying the two chain extenders. The properties of two elastomers were studied and the corresponding mechanism is discussed. In my opinion, the research is interesting, the scientific level of this manuscript is suitable to be published in Polymers as an article before consider more information provided.

  1. In the introduction part, although they are given in the abstract, the abbreviation that appears for the first time, such as TPU, PTMEG, BDM and BDH,etc. need to give the full name first.
  2. Line 125-127, why self-healing efficiency can be calculated by tensile strength, elongation and toughness??
  3. Figure 1 is not clear, please provide a clear picture.
  4. There are some English grammar mistakes in the article, please correct them.
  5. Figure 4 need to provide SEM of sample after healed.
  6. The style of Ref. is not right.

Author Response

Dear editor,

On behalf my co-authors, we would like to submit the revised manuscript entitled “Self-healable, strong and tough polyurethane elastomer enabled by carbamate containing chain extenders derived from ethyl carbonate” to your esteemed journal Polymers for publication. We have carefully done every single suggestion provided by reviewers, by either revising the manuscript or adding more detailed experimental results and discussion. Corresponding questions and replies are specifically presented one by one according to reviewers’ comments in the file of response to reviewers. We believe the quality of this revised manuscript has substantially improved and meets the demands of your journal Polymers for publication.

Yours sincerely

Yuhan Li, Ph.D.

Reviewer 2 Report

  1. Why did the authors choose BDH-2.0 and BDM-2.0 for FTIR characterization and then use BDH-1.7 and BDM-1.7 for the other characterizations? Please explain it.
  2. line 316 Te→The
  3. How did the authors determine the τ in Figure 5?
  4. Why not the samples were healed under the same temperature ranges for tensile test and adhesion test?
  5. elongation should be elongation at break in line 126.
  6. Unit for punctureenergy (mJ), please check mJ or MJ.

Round 2

Reviewer 2 Report

After checking the manuscript and the reply to comments, I think this manuscript could be accepted in the current form.